# Prolonged Subculturing of *Aspergillus fumigatus* on *Galleria* Extract Agar Results in Altered Virulence and Sensitivity to Antifungal Agents

**DOI:** 10.3390/cells12071065

**Published:** 2023-03-31

**Authors:** Aaron Curtis, Kieran Walshe, Kevin Kavanagh

**Affiliations:** Department of Biology, Maynooth University, W23 F2H6 Maynooth, Co. Kildare, Ireland

**Keywords:** *Aspergillus*, *Galleria*, infection, immunity, mini-model, passaging, virulence

## Abstract

*Aspergillus fumigatus* is an environmental saprophyte and opportunistic fungal pathogen of humans. The aim of the work presented here was to examine the effect of serially subculturing *A. fumigatus* on agar generated from *Galleria mellonella* larvae in order to characterize the alterations in the phenotypes that might occur. The passaged strains showed alterations in virulence, antifungal susceptibility, and in protein abundances that may indicate adaptation after 25 passages over 231 days on *Galleria* extract agar. Passaged strains demonstrated reduced virulence in *G. mellonella* larvae and increased tolerance to hemocyte-mediated killing, hydrogen peroxide, itraconazole, and amphotericin B. A label-free proteomic analysis of control and passaged *A. fumigatus* strains revealed a total of 3329 proteins, of which 1902 remained following filtration, and 32 proteins were statistically significant as well as differentially abundant. Proteins involved in the response to oxidative stress were altered in abundance in the passaged strain and included (S)-S-oxide reductase (+2.63-fold), developmental regulator FlbA (+2.27-fold), and histone H2A.Z (−1.82-fold). These results indicate that the prolonged subculturing of *A. fumigatus* on *Galleria* extract agar results in alterations in the susceptibility to antifungal agents and in the abundance of proteins associated with the oxidative stress response. The phenomenon may be a result of selection for survival in adverse conditions and highlight how *A. fumigatus* may adapt to tolerate the pulmonary immune response in cases of human infection.

## 1. Introduction

*Aspergillus fumigatus* is a ubiquitous soil-dwelling saprophyte and opportunistic fungal pathogen of humans [1]. *A. fumigatus* has been labeled as an ‘accidental’ pathogen primarily due to its independence of a host for survival, and its pathogenic potential may have evolved to facilitate survival in the environment [2]. Virulence factors that facilitate infection in mammals include siderophore secretion, the ability to grow at 37 °C, and the production of immunosuppressive toxins such as gliotoxin [3]. *A. fumigatus* can initiate growth within the host phagolysosome partially aided through the production of siderophores, and this process also occurs when *A. fumigatus* is engulfed by soil-dwelling amoebae [4]. Gliotoxin biosynthesis is thought to have evolved to combat free-living predatory amoebae during its saprophytic existence, posing a selection pressure on *A. fumigatus* [5]. These adaptations may contribute to the ability of the fungus to colonize and disseminate in human hosts.

In humans, aspergillosis can develop in neutropenic individuals, hematopoietic stem cell or solid organ transplant recipients, and patients on immunosuppressive therapy, and can manifest as a number of clinical conditions, ranging from allergic bronchopulmonary aspergillosis (ABPA) to acute invasive aspergillosis [6,7]. ABPA occurs as a result of hypersensitivity to *A. fumigatus* and affects 2–3.5% of patients with asthma [8] in addition to approximately 10.5% of cystic fibrosis (CF) patients, of which about 10% are chronically colonized [9,10]. Patients can be simultaneously colonized with different *A. fumigatus* strains, not all of which have the ability to persist in the pulmonary microenvironment [10]. A genotypic analysis demonstrated the persistence of *A. fumigatus* for at least 4.5 years within a CF patient, that persistent strains adapted to growth in hypoxic conditions, and that conidia were more sensitive to oxidative stress [11]. Repeat isolation from a single host with chronic granulomatous disease with persistent and recurrent invasive aspergillosis over two years revealed that the strains were isogenic and demonstrated resistance to itraconazole [12].

Chronic pulmonary aspergillosis requires prolonged antifungal therapy, with a recommended minimum course of 4–6 months [13]; therapy for up to 12 months may be required to improve long-term survival [14]. The long-term persistence of *A. fumigatus* in the lungs raises the possibility that the phenotype of the infecting strain may alter or adapt to the host microenvironment. The characterization of aspergillomas revealed resistance following antifungal therapy emerging from genetic alterations occurring within a fungal mass from a single parent strain [15]. An aspergilloma demonstrated an initial itraconazole MIC of 0.25 mg/L; after six months of antifungal therapy an MIC > 16 mg/L was evident, but after the cessation of therapy for four months the isolate MIC returned to 0.5 mg/L [16]. An analysis of *A. fumigatus* Af293 and CEA17, which share a 99.8% identical genome, demonstrated altered growth rates, virulence, and susceptibility to drug treatment and immune killing. The observed similarity in genomes and difference in phenotypes indicated that epigenetic alterations could be responsible for these physiological differences [17]. Another possible cause of these variations could be the presence of single-nucleotide polymorphisms, insertions, and deletions, which have been demonstrated to greatly impact heterogeneity [18]. 

The serial passaging of *A. fumigatus* on a murine lung homogenate medium revealed the selection of a rapidly germinating strain, after 13 passages, that produced an enhanced inflammatory response in mice. Genome sequencing revealed conserved mutations of the ssKA gene, which is part of the SakA mitogen-activated protein kinase (MAPK) stress pathway [19]. Serial passaging can also be conducted in vivo, and insects serve as an excellent model in which to passage as they are simple to maintain and inoculate, have short life cycles, and are easily manipulated [20]. *Galleria mellonella* larvae are a well-characterized model for studying bacterial and fungal pathogenesis due to the strong similarities between the insect immune response and the innate immune response of mammals in addition to the ability to grow at 37 °C [21,22]. Due to the similarities in the immune response, the preparation of agar from these larvae would contain products found in the human innate immune response, allowing for insights into how these products may shape fungal responses to the lung microenvironment. The serial passaging of *Cryptococcus neoformans* in *G. mellonella* larvae for 15 passages resulted in the generation of a distinct phenotype, which grew faster in hemolymph but was more susceptible to hydrogen peroxide in vitro, killed fewer murine macrophages, and produced a smaller fungal burden in human macrophages ex vivo compared to the parental strain [23]. Hemocytes exposed to the passaged strains produced less hydrogen peroxide, and a histopathological analysis also indicated that the passaged strain increased larval nodulation [23]. The serial passaging of *Aspergillus flavus* in *G. mellonella* larvae demonstrated that the genetic diversity of the passaged strain decreased significantly, which emphasizes the impact that the host exerts on shaping the evolution of a pathogen population in vivo [20].

*G. mellonella* larvae are susceptible to infection with *A. fumigatus*, and previously published results show a strong correlation with those obtained in mammals [24]. Larvae infected with *A. fumigatus* show many of the symptoms evident in infected mammals, including the development of granulomas and the in vivo production of toxins [25]. The aim of the work presented here was to characterize the effect of prolonged subculturing on *Galleria* extract agar on the virulence and antifungal response of *A. fumigatus*.

## 2. Materials and Methods

### 2.1. Aspergillus fumigatus Culture Conditions

*Aspergillus fumigatus* ATCC 26933 was cultured for 72 h at 37 °C on malt extract agar (MEA) (Oxoid, Basingstoke, UK) plates following point inoculation. Czapek–Dox broth (Duchefa Biochemie, Haarlam, The Netherlands) (50 mL) was inoculated with *A. fumigatus* conidia at an initial density of 1 × 10^5^ conidia/mL and grown at 37 °C for 72 h at 200 rpm in an orbital incubator. The wet biomass of mycelia was weighed at 72 h following filtration through Miracloth (Millipore, Millipore, MA, USA).

### 2.2. Generation of Passaged Strains of Aspergillus fumigatus

Gallerial extract agar (termed GEA20) was produced by grinding 20 *G. mellonella* larvae in 20 mL of sterile phosphate-buffered saline (PBS) via the use of a sterile mortar and pestle. The extract was centrifuged at 538× *g* for 5 min to remove particulate matter, and 20 mL of the supernatant was added to 80 mL of autoclaved agar (2 g *w*/*v*) supplemented with 0.1 g (*w*/*v*) of glucose, allowed to cool prior to addition, and 100 μL of penicillin–streptomycin (pen–strep) (Merck, Branchburg, NJ, USA) (10,000 U/10 mg/mL). *A. fumigatus* conidia were point inoculated onto GEA20 plates and incubated at 37 °C until growth reached the edge of the agar plate, which took an average of 9.24 days before subculturing onto fresh GEA20 plates. Three strains were selected after being serially passaged for a total of 25 passages over 231 days, and referred to as A25, C25, and E25. Control strains were serially sub-cultured on MEA plates for the same period of time. 

### 2.3. Virulence Assessment of Passaged Strains In Vivo

Six instar larvae of *G. mellonella* (Livefoods Direct Ltd., Sheffield, UK) were stored at 15 °C prior to use. Twelve larvae weighing 0.2–0.3 g, without signs of melanization, were inoculated with 20 μL of PBS containing 5 × 10^5^ control or passaged *A. fumigatus* conidia via intrahemocoel injection using a 26 G 1 mL syringe (Terumo, Tokyo, Japan). The larvae were placed in 9 cm Petri dishes and incubated at 37 °C. Larval viability was assessed over 72 h. Experiments were performed on four independent occasions.

### 2.4. Hemocyte Kill Assay

Hemolymph (500 µL) was extracted from *G. mellonella* larvae and hemocytes were harvested by centrifugation at 8609× *g* for 8 min. Cell-free hemolymph was retained on ice. Hemocytes were resuspended in 500 µL of sterile PBS and enumerated using a hemocytometer. The conidia of control and passaged *A. fumigatus* strains were harvested and resuspended in cell-free hemolymph for 30 min at 37 °C. The opsonized conidia were harvested and resuspended in 500 µL of sterile PBS. Conidial and hemocyte suspensions were mixed in a ratio of 1:1 (approximately 5 × 10^6^ hemocyte and conidia) in a final volume of 1 mL in a 50 mL Falcon tube (Sarstedt, Numbrecht, Germany) and incubated at 37 °C and 200 rpm. A 20 µL aliquot was taken at 20-minute intervals and serially diluted for plating on MEA plates in triplicate. Fungal colonies were enumerated to assess viability after incubation at 37 °C for 24 h. 

### 2.5. Susceptibility Testing

Hydrogen peroxide (Sigma, St. Louis, MO, USA) was serially diluted in Sabouraud dextrose broth (SDB) (Oxoid, Hampshire, UK) on a 96-well plate (Corning, Corning, NY, USA) to produce a concentration range between 30.6 and 245 mM. Amphotericin B (Sigma, St. Louis, MO, USA) and itraconazole (Sigma, St. Louis, MO, USA) were serially diluted in SDB, producing ranges of 0.78 to 62.5 mg/mL and 0.78 to 6.25 µg/mL, respectively. Conidia from the control and passaged strains were harvested and enumerated. Aliquots (100 μL) of conidia were added to each well of a 96-well plate (Corning, Corning, NY, USA) to provide a concentration of 1 × 10^5^ conidia per well. Plates were incubated at 37 °C and growth was assessed at 24 h at 600 nm using a plate reader (Bio-Tek Synergy HT, Somerset, NJ, USA).

### 2.6. Gliotoxin Extraction and Quantification

*A. fumigatus* cultures (*n* = 3) were grown in 50 mL of Czapek–Dox broth for 72 h. The supernatant was filtered through Miracloth, and 20 mL of supernatant was mixed 1:1 with chloroform for 2 h at room temperature. The chloroform fraction was stored at −20 °C overnight and samples were dried through rotary evaporation in a Büchi rotor evaporator (Brinkmann Instruments, Brea, CA, USA). Samples were dissolved in 500 µL of methanol and stored at −20 °C. Gliotoxin was detected by reverse phase HPLC (RP-HPLC; Shimadzu, Columbia, MD, USA). The mobile phase was 34.9% (*v*/*v*) acetonitrile (Fisher Scientific, Waltham, MA, USA), 0.1% (*v*/*v*) trifluoroacetic acid (TFA), (Sigma Aldrich, St Louis, MO, USA) and 65% (*v*/*v*) HPLC-grade water (ddH_2_O). Samples (20 µL) were loaded onto an Agilent ZORBAX SB-Aq 5 µm polar LC column and quantified based on the standard curve generated using gliotoxin standards dissolved in methanol ranging from 6.25 to 100 mg/mL. 

### 2.7. Total Secreted Siderophore Quantification

*A. fumigatus* cultures (*n* = 3) were grown in 50 mL of Czapek–Dox broth for 72 h. Siderophore activity in supernatants was determined via the use of a SideroTec HiSens assay (Accuplex, Maynooth, Ireland). Briefly, 100 μL of sample was added to a 96-well microplate followed by the addition of 100 μL of a ready-to-use detector. After 10 min of incubation at 37 °C, the plate was read on a fluorescent reader (360 excitation/460 emission). The siderophore concentration was determined by using desferoxamine as a reference standard.

### 2.8. Protein Extraction and Purification from A. fumigatus Hyphae

Protein extractions were performed as outlined previously [26]. *A. fumigatus* mycelia of control and passaged strain E25 (*n* = 3 per group) were grown for 72 h at 37 °C in Czapek–Dox media. Mycelium was harvested by filtration, snap-frozen in liquid nitrogen, and ground to a fine dust in a mortar via the use of a pestle. A lysis buffer (8 M urea, 2 M thiourea, and 0.1 M Tris-HCl (pH 8.0) dissolved in HPLC-grade ddH_2_O), supplemented with protease inhibitors (aprotinin, leupeptin, pepstatin A, tosyllysine chloromethyl ketone hydrochloride (TLCK) (10 µg/mL), and phenylmethylsulfonyl fluoride (PMSF) (1 mM/mL)), was added (4 mL/g of hyphae). The lysates were sonicated (Bandelin Senopuls), three times for 10 s at 50% power. The cell lysate was subjected to centrifugation (Eppendorf Centrifuge 5418) for 8 min at 14,500× *g* to pellet cellular debris. The protein concentration was quantified by the Bradford method and samples (100 µg) were subjected to overnight acetone precipitation. Samples were subjected to centrifugation at 14,500× *g* for 10 min to pellet proteins, acetone was removed, and the pellet was resuspended in a 25 µL sample resuspension buffer (8 M urea, 2 M thiourea, and 0.1 M Tris-HCl (pH 8.0) dissolved in HPLC-grade ddH_2_O). A 2 µL aliquot was removed from each sample for quantification via the Qubit quantification system (Invitrogen, Waltham, MA, USA). Ammonium bicarbonate (125 µL, 50 mM) was added to the remaining samples, which were subjected to reduction via the addition of 1 µL of 0.5 M dithiothreitol and incubated at 56 °C for 20 min, followed by alkylation with 0.55 M iodoacetamide at room temperature in the dark for 15 min. Proteins were digested via the addition of 1 µL of sequence-grade trypsin (Promega) (0.5 µg/µL), supplemented with 1 µL of Protease Max Surfactant Trypsin Enhancer (Promega 1% *w*/*v*), and incubated at 37 °C for 18 h. Digestion was quenched via the addition of 1 µL of TFA incubated at room temperature for 5 min. Samples were subjected to centrifugation at 14,500× *g* for 10 min prior to clean-up using C18 spin columns (Pierce). The eluted peptides were dried via the use of a SpeedyVac concentrator (Thermo Scientific (Waltham, MA, USA) Savant DNA120) and resuspended in 2% (*v*/*v*) acetonitrile and 0.05% (*v*/*v*) TFA aided by sonication for 5 min. The samples were centrifuged to pellet any debris at 14,500× *g* for 5 min, and 2 µL from each sample was loaded onto the mass spectrometer. 

### 2.9. Mass Spectrometry

Purified peptide extracts (2 μL containing 750 ng protein) were loaded onto a Q Exactive mass spectrometer (Thermo Fisher Scientific, Waltham, MA, USA) using a 133 min reverse-phase gradient, as per previous methods [21]. Raw MS/MS data files were processed through the Andromeda search engine in MaxQuant software v.1.6.3.4 110 using a *Neosartorya fumigata* reference proteome obtained from a UniProt-SWISS-PROT database to identify proteins (9647 entries, downloaded July 2022). The search parameters followed those described in [27].

### 2.10. Data Analysis

Perseus v.1.6.15.0 was used for the analysis, processing, and visualization of data. Normalized LFQ intensity values were used as the quantitative measurement of protein abundance. The data matrix generated was filtered to remove contaminants, and peptides were identified by site. LFQ intensity values were log2-transformed, and each sample was assigned to its corresponding group (control and E25). Proteins not found in all replicates in at least one group were omitted from further analysis. A data-imputation step was conducted to replace missing values with values that simulate signals of low-abundance proteins chosen randomly from a distribution specified by a downshift of 1.8 times the mean standard deviation of all measured values and a width of 0.3 times this standard deviation. Principle component analysis (PCA) was plotted using normalized intensity values. The proteins identified were then defined using a Perseus annotation file (downloaded in July 2022) to assign extract terms for biological process, molecular function, and Kyoto Encyclopedia of Genes and Genomes (KEGG) names. 

To visualize the differences between two samples, pairwise Student’s *t*-tests were performed using a cut-off of *p* < 0.05 on the post-imputation dataset. Volcano plots were generated by plotting the log2 fold change on the x-axis against the log *p*-values on the y-axis for each pairwise comparison. Statistically significant and differentially abundant (SSDA) proteins (ANOVA, *p* < 0.05) with a relative fold change greater than ±1.5 were retained for analysis. SSDA proteins were z-score normalized and then used for hierarchical clustering to produce a heat map. Identified SSDAs could then be assessed using Uniprot codes generated by Perseus to gain insights into their roles within the cells. The mass spectrometry proteomics data have been deposited to the ProteomeXchange Consortium via the PRIDE [28] partner repository with the dataset identifier PXD036787.

### 2.11. Statistical Analysis

Results from the phenotypic testing were assessed in Graphpad Prism Version 8.0.1. A 2-way ANOVA analysis or multiple paired *t*-tests were performed for the binary comparison of passaged and control *A. fumigatus* strains. Significance was set at *p* < 0.05. Proteomic analysis was conducted in Perseus V.1.6.15.0, as described above.

## 3. Results

### 3.1. Characterization of Growth Characteristics of Serially Subcultured A. fumigatus Strains

At the end of 25 passages, *A. fumigatus* conidia were isolated from GEA20 plates by washing with PBS/tween, and an aliquot of a diluted culture (5 × 10^6^/mL) was used to point inoculate MEA plates so that radial growth could be measured over 48 h at 37 °C. The results indicated no significant difference in the growth of the three passaged strains compared to the control (Appendix A). Conidia were also used to inoculate a flask of Czapex–Dox broth at a density of 1 × 10^5^/mL as well as incubated at 200 rpm and 37 °C for 72 h. At the end of the incubation period the mycelial wet biomass of the serially subcultured strains was not significantly different to that of the control (Appendix A). The gliotoxin concentration of culture filtrates at the end of 72 h was higher in the passaged strains than in the control, but only passaged strain A25 showed a significant result (*p* = 0.01) (Appendix A). There was no difference in the secreted siderophore concentrations between the control and passaged *A. fumigatus* strains (Appendix A). Pen–strep (0.1% *v*/*v*) was used in the GEA20 plates to prevent the overgrowth of bacteria from the *G. mellonella* digestive tract. The exposure of *A. fumigatus* to pen–strep did not affect the radial growth rate (Appendix A), and strains that were serially passaged on MEA agar plates containing 0.1% (*v*/*v*) pen–strep showed no significant alteration in their tolerance of hydrogen peroxide (Appendix A) or amphotericin B (Appendix A), but did demonstrate increased susceptibility to itraconazole (Appendix A), which was not observed in the GEA-passaged strains.

### 3.2. Passaged A. fumigatus Strains Show Reduced Virulence in G. mellonella Larvae

*G. mellonella* larvae were inoculated via an intrahemocoel injection with the conidia (1 × 10^5^/20 μL) of control or passaged *A. fumigatus* strains, and viability was assessed over 72 h (Figure 1). Infection with the control strain resulted in 30% mortality at 48 h compared to 5–8% mortality due to the passaged strains. By 72 h, the larvae infected with the control *A. fumigatus* conidia showed 44% mortality, compared to a mortality rate of 21–25% in the larvae infected with the conidia from the passaged strains (** *p* = 0.004 for A25 and C25; * *p* = 0.0106 for E25 relative to the control).

The response of the conidia from the control and passaged strains to *G. mellonella* hemocytes was assessed. Hemocytes were extracted from the larvae as described and mixed with hemolymph-opsonized conidia in a ratio of 1:1 at 37 °C. The viability in the conidia was assessed by serially diluting and plating them onto MEA plates. The results indicate that the hemocytes killed 92.8% of the control *A. fumigatus* conidia by 40 min and 98.04% by 100 min (Figure 2). In contrast, the conidia of the passaged strains were significantly less susceptible to hemocyte-mediated killing, demonstrating a 9.19–41.37% kill rate at 40 min and a 36.78–52.84% kill rate at 100 min (**** *p* < 0.0001 at both time points).

### 3.3. Passaged A. fumigatus Strains Show Altered Susceptibility to Antifungal Agents

In order to confirm the toleration of oxidative stress indicated in Figure 2, the susceptibility of the conidia of control and passaged A. fumigatus strains to hydrogen peroxide, as well as the antifungal agents amphotericin B and itraconazole, was assessed as described. The passaged strains showed increased growth in the presence of hydrogen peroxide compared to the control A. fumigatus strain at concentrations from 30.62 to 245 mM, with a significant increase in the growth of the E25 strain at concentrations of 30.62, 61.25, and 245 mM (**** *p* < 0.0001) (Figure 3A). All passaged strains demonstrated significantly increased growth at amphotericin B concentrations of 0.78–3.125 mg/mL compared to the control (**** *p* < 0.0001) (Figure 3B). The passaged strains showed significantly increased growth at all of the itraconazole concentrations tested (Figure 3C). 

### 3.4. Proteomic Characterization of Passaged A. fumigatus Strain E25

A quantitative proteomic analysis was employed to identify alterations in the proteome of passaged strain E25 that might explain the altered susceptibilities to hemocyte-mediated killing and the antifungal agents described above. In total, 3329 proteins were detected, of which 1902 remained following filtering. Thirty-two proteins were statistically significant and differentially abundant (SSDA) (Table 1). 

The heat map of proteins altered in abundance indicates a clear difference between the control and passaged strains (Figure 4). Proteins such as polyadenylation factor subunit CstF64 (+3.58-fold) and nuclear cap-binding protein subunit 2 (+2.47-fold), involved in mRNA stability, were increased in abundance in the passaged strain. Peptide-methionine (S)-S-oxide reductase (+2.63-fold) and developmental regulator FlbA (+2.27-fold), involved in the response to oxidative stress, were also increased in abundance in the passaged strains. Proteins decreased in abundance in the passaged strain included zinc/cadmium resistance protein (−5.17-fold), translation initiation regulator (Gcn20) (−1.96-fold), and glucose repressible protein Grg1 (−1.87-fold). In addition, nucleoporin NUP49/NSP49 (−2.24-fold), guanyl-nucleotide exchange factor (Sec7) (−1.76-fold), and histone H2A.Z (−1.82-fold) were also decreased in abundance (Figure 5). The results indicate that passaged strain E25 showed some alterations in the proteome, and those that are present may contribute to the increased tolerance to oxidative stress. 

## 4. Discussion

To examine the factors facilitating the selection and persistence of *A. fumigatus* in a host, in vivo or in vitro serially passaging can be employed to impose specific conditions for prolonged periods of time. The selective serial culturing of an organism can result in genetic modifications as the organism responds to a given environmental pressure [20]. To facilitate prolonged passaging and overcome the relatively short lifespan of *G. mellonella* larvae, an agar system was formulated. The results presented show that the prolonged subculturing of *A. fumigatus* on GEA20 plates selected strains showing reduced virulence in *G. mellonella* larvae but with greater tolerance of hemocyte-mediated killing as well as hydrogen peroxide, amphotericin B, and itraconazole. Reduced virulence was not a result of altered growth, as control and passaged strains grew at the same rate (Appendix A). In addition, only one passaged strain demonstrated significant alterations in gliotoxin production (Appendix A), and there was no significant alteration in siderophore secretion (Appendix A). A small number of changes in the proteome was detected in passaged strain E25 (Table 1), and those proteins increased in abundance, such as peptide-methionine (S)-S-oxide reductase (+2.63-fold) and developmental regulator FlbA (+2.27-fold), coupled with the reduced abundance of histone H2A.Z (−1.82-fold) could have contributed to the observed tolerance of oxidative stress. FlbA negatively affects the responses of detoxification to reactive oxygen species (ROS) as well as gliotoxin, and negatively regulates GliT expression. The absence of FlbA increased ROS accumulation in hyphae, which elevates the expression of ROS scavengers such as catalase and superoxide dismutase [29]. H2A.Z has been found to be involved in genome stability, DNA repair, and transcriptional regulation across eukaryotes. In *Neurospora crassa*, H2A.Z regulates the oxidative stress response [30]. H2A.Z antagonizes CPC1 binding to restrict cat-3 expression in a normal setting, whereas under oxidative stress H2A.Z is removed from chromatin, leading to a rapid and full activation of cat-3 transcription, enhancing the capacity of resistance to physiological stimuli [31]. Proteomic evidence also indicates that these alterations have not arisen as a result of starvation or nutrient deprivation, as proteins involved in the starvation response, such as zinc/cadmium resistance protein (−5.17-fold), expressed in low-iron environments [32], and glucose repressible protein grg1 (−1.81-fold), which is expressed in low-nutrient environments [33], were reduced in abundance. 

The data presented here indicate that the prolonged subculturing of *A. fumigatus* on GEA20 plates resulted in the selection of phenotypically fit variants that could persist in the culture conditions and withstand oxidative stress. The proteomic analysis of the passaged and control strains indicated alterations primarily involved in mRNA stability and oxidative stress tolerance. The results suggest that culture conditions may serve as a selective bottleneck, as previously demonstrated in work conducted with *A. flavus* [20]. In addition, *C. neoformans* strains passaged in *G. mellonella* larvae demonstrated increased oxidative stress tolerance by downregulating hydrogen peroxide production via the shedding of the immunomodulatory capsule [23]. In the work presented here, the passaged *A. fumigatus* strains also demonstrated increased tolerance to oxidative stress, further emphasizing the importance of these mechanisms. Itraconazole and amphotericin B, which target the fungal cell membrane, affect fungal redox homeostasis by increasing intracellular ROS production. These responses were abolished via the inhibition of mitochondrial respiratory complex I, which suggests that mitochondrial complex I is the main source of deleterious ROS production in *A. fumigatus* challenged with antifungal compounds [34]. Interestingly, the proteomic analysis indicated an increased expression of complex I intermediate associated protein (+3.44-fold) independent of exposure to any antifungal agent. Previous work has shown that the exposure of *Candida albicans* to hydrogen peroxide for 60 min increased tolerance to caspofungin through the simultaneous activation of the Cap and Hog pathways [35], indicating that prior exposure to elevated internal ROS could be attributed to reduced susceptibility to amphotericin B and itraconazole. The production of ROS by the innate immune response is crucial to protection against colonization. The ability of *A. fumigatus* to adapt and persist in the presence of these stressors and innate immune products could play a role in influencing antifungal susceptibility and response to the immune cells in vivo. 

## 5. Conclusions

The data presented here suggest that prolonged subculturing on GEA20 plates can alter the phenotype and proteome of *A. fumigatus*. The passaged strains demonstrated reduced virulence in vivo and increased tolerance to hemocyte-mediated killing, hydrogen peroxide, itraconazole, and amphotericin B. This tolerance may be due to the proteomic alterations evident in the passaged strains conferring tolerance to oxidative stress. Prolonged *A. fumigatus* colonization in vivo may also lead to strains better adapted to the pulmonary environment as well as those that display enhanced tolerance to antifungal agents. Such a process may have implications for human health, as inadvertent selection for drug-tolerant and persistent strains could complicate therapy. 

## Figures and Tables

**Figure 1 cells-12-01065-f001:**
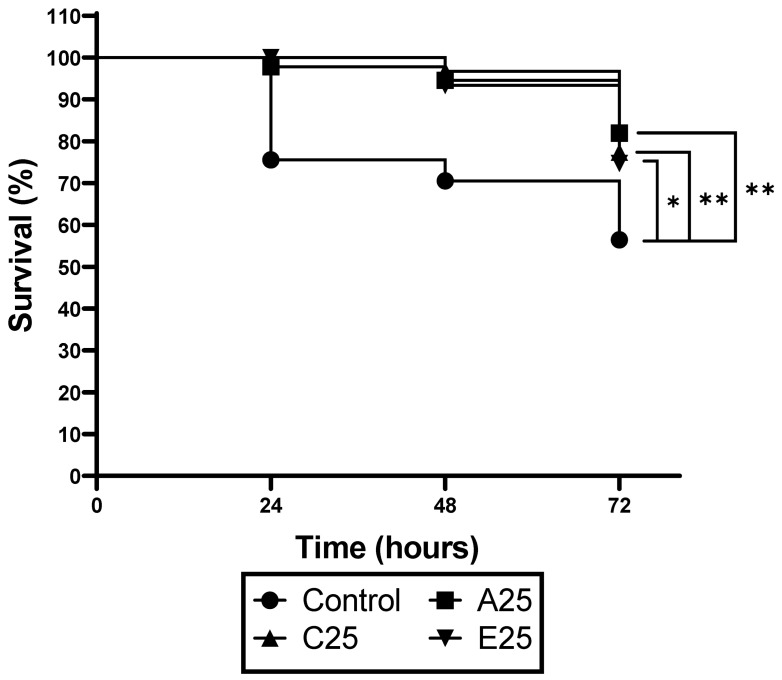
Response of larvae to infection by the conidia of control and passaged *A. fumigatus* strains. Larvae were infected with the conidia as described, and their survival was monitored over 72 h. Passaged strains showed significantly reduced virulence at 72 h (A25, * *p* = 0.0458; C25 and E25, ** *p* = 0.042. Logrank (Mantel–Cox) test).

**Figure 2 cells-12-01065-f002:**
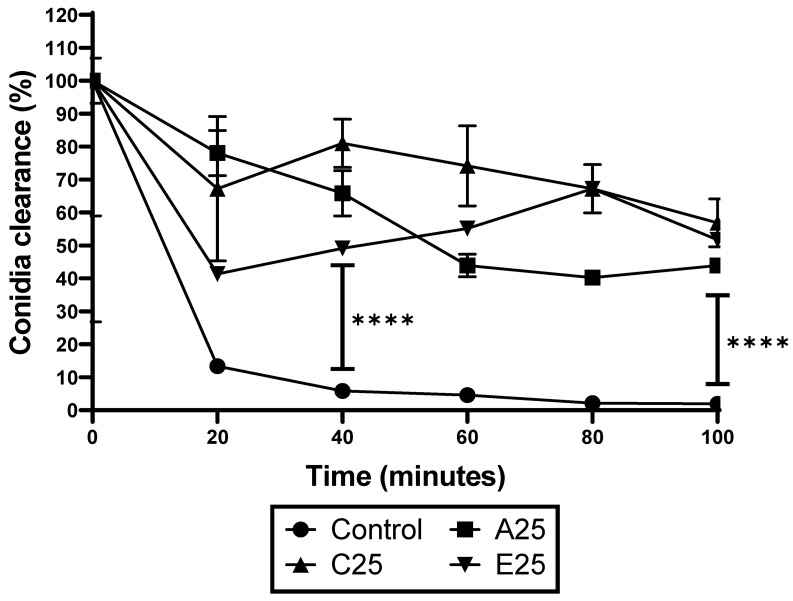
Response of *A. fumigatus* conidia to hemocyte-mediated killing. The conidia of A. *fumigatus* passaged and control strains were exposed to *G. mellonella* hemocytes ex vivo. The passaged strains demonstrated significantly increased tolerance to immune cell killing at 40 min (**** *p* < 0.0001) and 100 min (**** *p* < 0.0001), determined by a one-way ANOVA followed by pair-wise multiple comparisons using the Tukey test.

**Figure 3 cells-12-01065-f003:**
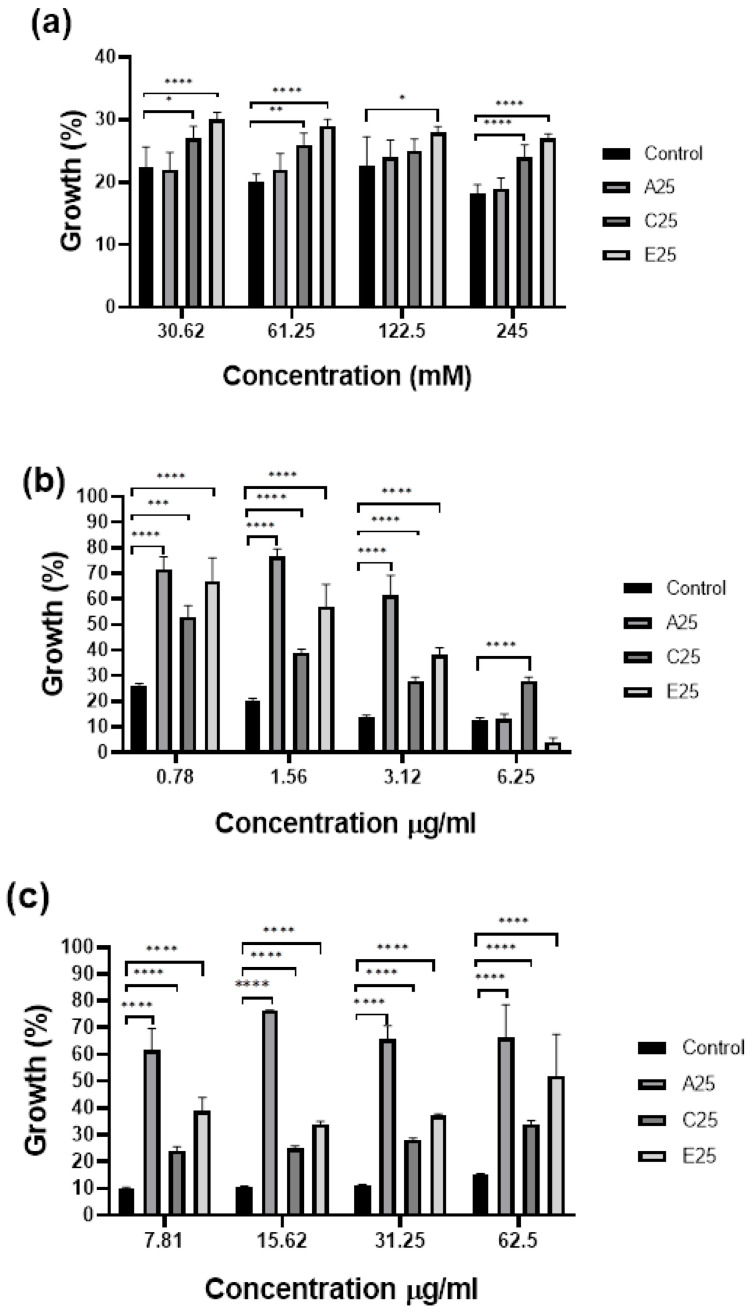
Analysis of the response of control and passaged *A. fumigatus* conidia to hydrogen peroxide, amphotericin B, and itraconazole. Passaged *A. fumigatus* conidia demonstrated significantly increased tolerance to (**a**) hydrogen peroxide (strain E25 at concentrations of 490, 122.5, and 61.25 mM (**** *p* < 0.0001)); (**b**) amphotericin B at concentrations of 3.12–0.78 mg/mL in all of the passaged strains compared to the control (**** *p* < 0.0001); and (**c**) itraconazole in all of the passaged strains compared to the control with C25 at 56.25 µg/mL (*** *p* = 0.0017) and all other strains as well as concentrations (**** *p* < 0.0001) determined by a one-way ANOVA followed by pair-wise multiple comparisons using the Tukey test for each treatment dose (* *p* = 0.01, ** *p* = 0.001).

**Figure 4 cells-12-01065-f004:**
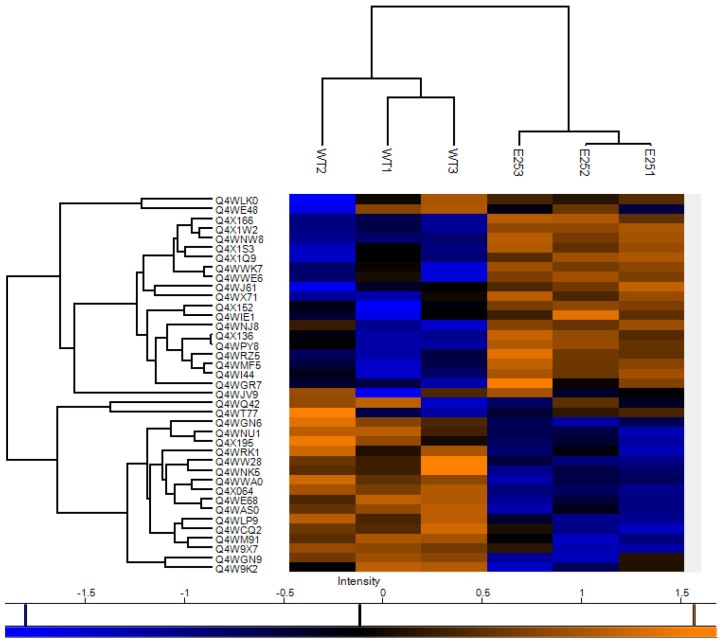
Heat map showing protein clustering between control and passaged *A. fumigatus* strain E25. Shotgun quantitative proteomic analysis of passaged strain and control mycelia grown in Czapex–Dox broth for 72 h. Two-way unsupervised hierarchical clustering of the median protein expression values of all statistically significant differentially abundant proteins. Hierarchical clustering (columns) identified two distinct clusters comprising the three replicates from their original sample groups.

**Figure 5 cells-12-01065-f005:**
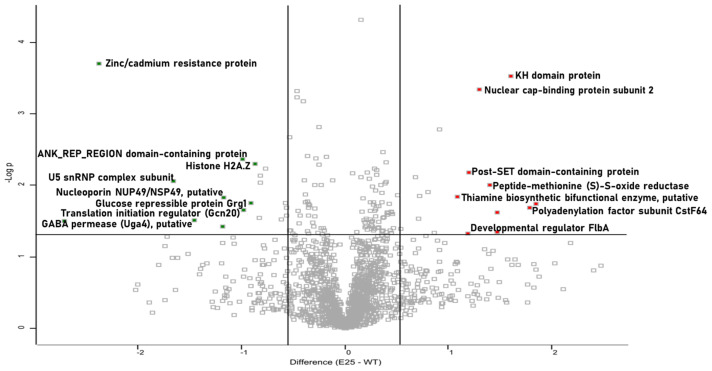
Volcano plots showing alterations in protein abundance in control and passaged *A. fumigatus* proteomes. Volcano plots of all identified proteins based on relative abundance differences between passaged strains and control mycelia. Volcano plots showing the distribution of quantified proteins. Proteins above the line are considered statistically significant (*p* value < 0.05), and those to the right and left of the vertical lines indicate relative fold changes greater than 1.5-fold.

**Table 1 cells-12-01065-t001:** All proteins detected to be statistically significant and differentially abundant in passaged strain E25 relative to the control *A. fumigatus* strain.

Protein Name	Gene Name	Peptides	Sequence Coverage (%)	Score	Fold Change
Polyadenylation factor subunit CstF64, putative	AFUA_2G09100	3	18.9	17.95	3.58
Complex I intermediate associated protein (Cia30), putative	AFUA_3G06220	4	15.2	14.82	3.44
KH domain protein	AFUA_4G07220	6	19.1	22.51	3.04
Uncharacterized protein	AFUA_3G08440	3	22.6	9.074	2.77
50S ribosomal protein L3	AFUA_4G06000	4	20.5	19.30	2.77
Peptide-methionine (S)-S-oxide reductase	AFUA_2G03140	3	49.1	26.35	2.63
Nuclear cap-binding protein subunit 2	AFUA_2G08570	5	25	7.55	2.46
Post-SET domain-containing protein	AFUA_6G10080	2	23.3	8.33	2.28
Developmental regulator FlbA	AFUA_2G11180	4	7.4	10.65	2.27
Thiamine biosynthetic bifunctional enz, putative	AFUA_2G08970	8	27.2	19.26	2.11
RING-type E3 ubiquitin transferase	AFUA_2G11040	4	29.2	6.07	1.88
Small nuclear ribonucleoprotein E	AFUA_7G05980	4	32.6	9.08	1.81
Phosphatidylglycerol/phosphatidylinositol transfer protein	npc2	10	35.4	29.44	1.74
DlpA domain protein	AFUA_4G10940	5	23.4	15.65	1.64
RSC complex subunit (RSC1), putative	AFUA_3G05560	4	7.8	9.43	1.63
Probable mannosyl-oligosaccharide alpha-1,2-mannosidase 1B	mns1B	14	47.5	64.36	1.60
ATP-dependent RNA helicase dbp2	dbp2	17	43	109.23	−1.53
Rhomboid protein 2	rbd2	2	15.4	19.36	−1.69
Integral ER membrane protein Scs2, putative	AFUA_4G06950	9	40.2	53.75	−1.75
Translation elongation factor eEF-3, putative	AFUA_7G03660	68	77.1	323.31	−1.76
Guanyl-nucleotide exchange factor (Sec7), putative	AFUA_7G05700	8	5.8	27.80	−1.76
Histone H2A.Z	htz1	6	51.4	65.54	−1.82
Glucose repressible protein Grg1, putative	AFUA_5G14210	2	34.8	68.31	−1.87
Translation initiation regulator (Gcn20), putative	AFUA_4G06070	9	20.2	37.21	−1.96
ANK_REP_REGION domain-containing protein	AFUA_5G14930	20	48	190.54	−1.98
Nucleoporin NUP49/NSP49, putative	AFUA_6G10730	4	10.8	10.05	−2.24
Aminotransferase, putative	AFUA_6G02030	6	23	16.58	−2.26
GABA permease (Uga4), putative	AFUA_4G03370	4	8.9	12.47	−2.72
U5 snRNP complex subunit, putative	AFUA_7G02280	7	32.6	19.75	−3.14
Zinc/cadmium resistance protein	AFUA_2G14570	1	3	9.01	−5.17
Uncharacterized protein	AFUA_1G16030	16	38.2	62.94	−6.54

## Data Availability

The mass spectrometry proteomics data have been deposited to the ProteomeXchange Consortium via the PRIDE [22] partner repository, with the dataset identifier PXD036787.

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
