# Peer review of "Prolonged Subculturing of *Aspergillus fumigatus* on *Galleria* Extract Agar Results in Altered Virulence and Sensitivity to Antifungal Agents"

_cells, 2023, doi:10.3390/cells12071065_

Round 1
Reviewer 1 Report
I congratulate the authors for an interesting study.
Evaluating and interpreting it correctly must provide clear, constructive insights - what was done, why, and what was new. In addition, the presentation of results should be of high quality (correct names of x, y-axis and etc.). Otherwise, the work loses some of its value and looks unprofessional.
So, after the corrections, the manuscript can be published.
Abstract
1. What is the mean ''32 proteins were statistically significant and differentially abundant in 16 line?
2. The aim of work should be presented in the abstract.
Introduction:
3. The last section of the introduction should reveal the problem and the importance of your research. What does this article bring to scientific knowledge? Could you write a more focused section?
Materials and methods:
4. The countries of used chemicals and equipment companies should be indicated, For example: (Oxoid, UK).
5. 80 line: milliliters should be abbreviated ml, not mls – rewrite in all methods section.
6. 81 line: ‘’grown at 37°C for 72h at 200 rpm‘‘ – grown where? How did you measure the weight of mycelia?
7. 86, 102,149 lines: should be rewritten: the speed of centrifugation x g, for example, 535 x g.
8. 108 line: what is mean ‘’universal’?
Results:
9. Consistency about figures not maintained. For example Fig. 1 in 203 line and Fig 1 (without dot) in 212-215 lines. Furthermore, in the journal requirements, the reference to figures should be written as follows: (Figure 1).
10. The journal requires that figures a, b should be written in lowercase letters in parentheses and the middle of the figure at the bottom. See: https://www.mdpi.com/journal/cells/instructions#oriimages
11. Figure 1: Y axis should be rewritten: Survival (%), because uniformity with the X-axis must be maintained – correct in all figures. Also, ‘’wild-type’’, instead of ‘’wildtype’’ – correct in all figures.
12. The section on the results should describe only results without their interpretations. However, interpretations should be written in the discussion section. For example – 285-287 lines should be in the discussion.
13. 268 line: What does the statement: Thirty-two proteins were statistically significant differentially abundant (SSDAs) (Table 1)“ mean?
Disscusion:
14. Include in the discussion the limitation of the study.
15. The first and second paragraphs are more suitable for introduction, not for discussion if authors only describe the results from other publications, not the results from the current work.
16. Generally speaking, the discussion should be structured as follows: Authors present their obtained results and how they relate to the results of other researchers. Then follow the explanation of the differences between the authors' results and those of other researchers.
17. I missed a clear discussion in the work of authors themselves. It is not highlighted what is obtained by the authors and what is a citation of other publications. Write the discussion in a more structured way.
Conclusions
18. I recommend to include the conclusion of your research.
References:
19. Please correct according to the requirements of the journal:
1. Author 1, A.B.; Author 2, C.D. Title of the article. Abbreviated Journal Name Year, Volume, page range.
Author Response
Reviewer 1
I congratulate the authors for an interesting study.
Evaluating and interpreting it correctly must provide clear, constructive insights - what was done, why, and what was new. In addition, the presentation of results should be of high quality (correct names of x, y-axis and etc.). Otherwise, the work loses some of its value and looks unprofessional.
So, after the corrections, the manuscript can be published.
Abstract
- What is the mean ''32 proteins were statistically significant and differentially abundant in 16 line?
Response: The 32 proteins refers to proteins that remained following statistical analysis and thus they were the key focus for the study. These proteins were significantly different in their abundance when compared to a control sample.
- The aim of work should be presented in the abstract.
Response: Thank you for your suggestion, this has been edited in text see line 12 and 13
Introduction:
- The last section of the introduction should reveal the problem and the importance of your research. What does this article bring to scientific knowledge? Could you write a more focused section?
Response: Thank you for this observation please see the changes in the text lines 91 to 96
Materials and methods:
- The countries of used chemicals and equipment companies should be indicated, For example: (Oxoid, UK).
Response: Thank you, these changes have been made in the text
- 80 line: milliliters should be abbreviated ml, not mls – rewrite in all methods section.
Response: Thank you, this has been changed in the text
- 81 line: ‘’grown at 37°C for 72h at 200 rpm‘‘ – grown where? How did you measure the weight of mycelia?
Response: Edited in text to clarify see line 109 and 110.
- 86, 102,149 lines: should be rewritten: the speed of centrifugation x g, for example, 535 x g.
Response: Thank you for this observation this has been edited throughout the text
- 108 line: what is mean ‘’universal’?
Response: Edited for clarification please see line 141
Results:
- Consistency about figures not maintained. For example Fig. 1 in 203 line and Fig 1 (without dot) in 212-215 lines. Furthermore, in the journal requirements, the reference to figures should be written as follows: (Figure 1).
Response: Thank you, for your observation the figures have been made consistent and to the correct format throughout the text
- The journal requires that figures a, b should be written in lowercase letters in parentheses and the middle of the figure at the bottom. See: https://www.mdpi.com/journal/cells/instructions#oriimages
- Figure 1: Y axis should be rewritten: Survival (%), because uniformity with the X-axis must be maintained – correct in all figures. Also, ‘’wild-type’’, instead of ‘’wildtype’’ – correct in all figures.
Response: The axis of the graphs have been altered and are now consistent
- The section on the results should describe only results without their interpretations. However, interpretations should be written in the discussion section. For example – 285-287 lines should be in the discussion.
Response: We would like to leave this text here as it provides some information on the significance of the results, before being discussed in the Discussion.
- 268 line: What does the statement: Thirty-two proteins were statistically significant differentially abundant (SSDAs) (Table 1)“ mean?
Response: The 32 proteins refers to proteins that remained following statistical analysis and thus they were the key focus for the study. These proteins were significantly different In their rate of expression when compared to a control sample.
Disscusion:
- Include in the discussion the limitation of the study.
Response: Added in the Discussion
- The first and second paragraphs are more suitable for introduction, not for discussion if authors only describe the results from other publications, not the results from the current work.
Response: These paragraphs have been moved to the introduction of the paper (see lines 56-71).
- Generally speaking, the discussion should be structured as follows: Authors present their obtained results and how they relate to the results of other researchers. Then follow the explanation of the differences between the authors' results and those of other researchers.
Response: We have redrafted the Discussion in accordance with these suggestions
- I missed a clear discussion in the work of authors themselves. It is not highlighted what is obtained by the authors and what is a citation of other publications. Write the discussion in a more structured way.
Response: Completed in text
Conclusions
- I recommend to include the conclusion of your research.
Response: A conclusion has been added to the paper (lines 401- 410).
References:
- Please correct according to the requirements of the journal:
1. Author 1, A.B.; Author 2, C.D. Title of the article. Abbreviated Journal NameYear, Volume, page range.
Response: All reference are now in the correct format for the Journal.
Reviewer 2 Report
It is suggested to improve the clarity of figures 1, 2, 3 A-C and 5; especially, it is suggested to standardize the width and thickness of the axes, size of the texts, intervals between marks, design of symbols and box fillings; as well as to reduce the overlap between adjacent symbols. Also, it is convenient to specify in the figure caption the synonymy between the wildtype and the control strains of A. fumigatus (as in the materials and methods section of the manuscript).
It is suggested to include in the materials and methods section of the manuscript all statistical tests shown in the results to visualize the differences detected between samples.
Author Response
Reviewer 2
it is suggested to improve the clarity of figures 1, 2, 3 A-C and 5; especially, it is suggested to standardize the width and thickness of the axes, size of the texts, intervals between marks, design of symbols and box fillings; as well as to reduce the overlap between adjacent symbols. Also, it is convenient to specify in the figure caption the synonymy between the wildtype and the control strains of A. fumigatus (as in the materials and methods section of the manuscript).
Response: Figures have been standardised to the best of the authors abilities thank you for your suggestions. Symbols and intervals have been standardised.
It is suggested to include in the materials and methods section of the manuscript all statistical tests shown in the results to visualize the differences detected between samples.
Response: Please see added section lines 235-239
Reviewer 3 Report
The current manuscript focus on the differences between fungal strains subcultured for 25 times, versus a control which is supposed to have been subcultured only once. It is not made clear what is the relevance of this study. Moreover, the discussion would be more interesting if the methods and choices made were explained and justified, and if the results obtained were compared with published references and discussed. Part of the discussion reads more as an introduction. Therefore, some issues should be addressed.
Please find below the detailed comments and suggestions.
Line 1: Make sure that in the title as well as through the entire document all scientific names are italicised. Also, titles do not have punctuation. Please delete the full stop.
Lines 10-12: The sentence written is not very clear. Did you use larvae to produce the fungi? Please revise.
Line 20: Were the alterations noted found on the “oxidative stress” or on the “susceptibility to oxidative stress”? Please revise if necessary.
Lines 22-23: Do you mean “adapt and survive”? Or “adapt to survive”? Or do you mean “survive”? Please revise.
Lines 32-34: You are stating that siderophores produced are the reason for the fungus to start growing. Is this what you really want to say? Please revise if necessary.
Line 45: Please replace the word “isolates” with the word “strains”.
Line 47: Please replace the word “sufferer” with the word “patient”.
Line 55: Please write the MAPK in full before the acronym, and explain what is the relevance of focusing on this pathway.
Line 70: To which results are you referring to? Please clarify this.
Line 80: Where you have written the word “density”, I suggest writing “concentration” instead.
Line 81: Consider writing “Mycelial wet biomass was weighted…”.
Line 82: There is a full stop missing at the end of the sentence.
Line 86: The term “hand-hot” is not the most suitable. Consider rewriting. Also, where you wrote “… and 20 mls of the supernatant was added…”, consider writing “… and 20 ml of the supernatant were added…”.
Line 88: Please write “penstrep” in full.
Line 89: Why did you grow your samples for a period of 9.24 and not a more common number of days, e.g., 7?
Line 90: When you say that “the strains were produced”, do you mean “selected”? If so, please revise accordingly.
Line 98: Note that “Petri” should have a capital first letter since it this comes from the name of a person. Also, when you mention “… over 72h.”, do you mean “… after 72h.”? If so, please revise.
Line 108: What are you refereeing to when you write “Universal”?
Line 122: When you mention ml, this is usually a plural amount, therefore, instead of writing “was mixed” you should write “were mixed”.
Line 128: The number 2 in H2O should be subscript.
Line 140: Why did you only used strain E25?
Lines 143, 152: You have already used ddH2O as HPLC-grade water in line 128. Please be consistent and when referring to this, write ddH2O.
Line 144: Please write all acronyms in full when used for the first time in the text. Where you have written “was added”, you should write “were added”.
Line 164: Instead of “ul”, please write “µl”.
Line 270: In table 1, one of the rows is highlighted. Is this on purpose? If so, explain this in the legend.
Lines 302-321: This is introductory information. Consider moving this to the introduction section of your manuscript.
Line 328: It would be interesting to read here about the reasons for screening for hydrogen peroxide, amphotericin B, and itraconazole.
Line 329: It has not been made clear why at a certain point of your methodology you excluded some strains and did most of the work with only one strain.
Line 333: Write the full name of ROS at its first mention.
Lines 351-354: This has been mentioned before in the introduction. Here it would be more interesting to discuss the relation of this information with your own work.
Line 365: Use the acronym ROS in this sentence.
Author Response
Reviewer 3
The current manuscript focus on the differences between fungal strains subcultured for 25 times, versus a control which is supposed to have been subcultured only once. It is not made clear what is the relevance of this study. Moreover, the discussion would be more interesting if the methods and choices made were explained and justified, and if the results obtained were compared with published references and discussed. Part of the discussion reads more as an introduction. Therefore, some issues should be addressed.
Response: Please note the passaged strains were sub-cultured on Galleria extract agar for 25 passages, the control strain was sub-cultured on Malt Extract Agar for an equivalent number of passages.
Please find below the detailed comments and suggestions.
Line 1: Make sure that in the title as well as through the entire document all scientific names are italicised. Also, titles do not have punctuation. Please delete the full stop.
Response: This has been corrected in text
Lines 10-12: The sentence written is not very clear. Did you use larvae to produce the fungi? Please revise.
Response: Edited to clarify that the agar was made form larval tissue
Line 20: Were the alterations noted found on the “oxidative stress” or on the “susceptibility to oxidative stress”? Please revise if necessary.
Response: Edited in text to clarify.
Lines 22-23: Do you mean “adapt and survive”? Or “adapt to survive”? Or do you mean “survive”? Please revise.
Response: Edited in text to clarify.
Lines 32-34: You are stating that siderophores produced are the reason for the fungus to start growing. Is this what you really want to say? Please revise if necessary.
Response: Edited to state siderophores only aid in growth initiation
Line 45: Please replace the word “isolates” with the word “strains”.
Response: Edited throughout the text to clarify.
Line 47: Please replace the word “sufferer” with the word “patient”.
Edited in text to clarify.
Line 55: Please write the MAPK in full before the acronym, and explain what is the relevance of focusing on this pathway.
Response: Clarified in text
Line 70: To which results are you referring to? Please clarify this.
Response: Edited in text to clarify.
Line 80: Where you have written the word “density”, I suggest writing “concentration” instead.
Response: When referring to number of cell/ml we use the word ‘density’ when referring to the amount of a chemical dissolved in a liquid we use the word ‘concentration’
Line 81: Consider writing “Mycelial wet biomass was weighted…”.
Response: Edited in text to clarify
Line 82: There is a full stop missing at the end of the sentence.
Response: Edited in text to clarify
Line 86: The term “hand-hot” is not the most suitable. Consider rewriting. Also, where you wrote “… and 20 mls of the supernatant was added…”, consider writing “… and 20 ml of the supernatant were added…”.
Response: Edited in text to clarify
Line 88: Please write “penstrep” in full.
Response: Edited in text to clarify
Line 89: Why did you grow your samples for a period of 9.24 and not a more common number of days, e.g., 7?
Response: Edited in text to clarify – this was the average length of each subculturing
Line 90: When you say that “the strains were produced”, do you mean “selected”? If so, please revise accordingly.
Response: This has been revised in the text.
Line 98: Note that “Petri” should have a capital first letter since it this comes from the name of a person. Also, when you mention “… over 72h.”, do you mean “… after 72h.”? If so, please revise.
Response: Edited in text to clarify
Line 108: What are you refereeing to when you write “Universal”?
Response: Edited in text to clarify
Line 122: When you mention ml, this is usually a plural amount, therefore, instead of writing “was mixed” you should write “were mixed”.
Response: Edited in text to clarify
Line 128: The number 2 in H2O should be subscript.
Response: Corrected in text to
Line 140: Why did you only used strain E25?
Response: Due to financial limitations only one strain was assessed using proteomic analysis
Lines 143, 152: You have already used ddH2O as HPLC-grade water in line 128. Please be consistent and when referring to this, write ddH2O.
Response: Edited in text to clarify
Line 144: Please write all acronyms in full when used for the first time in the text. Where you have written “was added”, you should write “were added”.
Response: Edited in text to clarify
Line 164: Instead of “ul”, please write “µl”.
Response: Edited in text to clarify
Line 270: In table 1, one of the rows is highlighted. Is this on purpose? If so, explain this in the legend
Response: This row is the title row of the graph .
Lines 302-321: This is introductory information. Consider moving this to the introduction section of your manuscript.
Response: This material has been moved to the introduction as suggested
Line 328: It would be interesting to read here about the reasons for screening for hydrogen peroxide, amphotericin B, and itraconazole.
Response: Clarified in text please see line 370
Line 329: It has not been made clear why at a certain point of your methodology you excluded some strains and did most of the work with only one strain.
Response: The high cost of proteomic analysis forced us to use only one of the sub-cultured strains for this part of the work.
Line 333: Write the full name of ROS at its first mention.
Response: Edited in text to clarify
Lines 351-354: This has been mentioned before in the introduction. Here it would be more interesting to discuss the relation of this information with your own work.
Response: Clarification added see Lines 384-388
Line 365: Use the acronym ROS in this sentence.
Response: Edited in text
Round 2
Reviewer 3 Report
The majority of the previous comments was answered and most suggestions were accepted which greatly improved the manuscript.
However, the following issues were not resolved:
Line 55: Please write the MAPK in full before the acronym, and explain what is the relevance of focusing on this pathway.
Response: Clarified in text
You have not explained the relevance of focusing on the MAPK, please revise.
Line 70: To which results are you referring to? Please clarify this.
Response: Edited in text to clarify.
There is no edited text. The only change was the format of the colour of the text which is now in red. Please clarify what are the results being mentioned in lines 94-95.
Line 81: Consider writing “Mycelial wet biomass was weighted…”.
Response: Edited in text to clarify
Now in line 106, where you have “Mycelia wet weights were weighed at…” I suggest again writing “Mycelial wet biomass was weighted…”.
Line 90: When you say that “the strains were produced”, do you mean “selected”? If so, please revise accordingly.
Response: This has been revised in the text.
This is still as it was in the previous version, only now it is in lines 117-118.
Line 328: It would be interesting to read here about the reasons for screening for hydrogen peroxide, amphotericin B, and itraconazole.
Response: Clarified in text please see line 370
There is no clarification regarding the previous comment in line 370.
Author Response
Author comment: We are very grateful for the comments of the reviewer and have addressed these in the manuscript to the best of our ability. I am very sorry we did not adequately address these in the previous version which was partly due to the very large number of edits we had to introduce into the previous version of the manuscript (52). Thank you for your helpful comments on our work.
Comments attached
